# Synthesis and Processing of Near Infrared—Activated Vitrimer Nanocomposite Films Modified with β-Hydroxyester-Functionalized Multi-Walled Carbon Nanotubes

Tomás E. Byrne Prudente [1], Diandra Mauro [2], Julieta Puig [1,*], Facundo I. Altuna [1,*], Tatiana Da Ros [2] and Cristina E. Hoppe [1,3]

1 Instituto de Investigaciones en Ciencia y Tecnología de Materiales (INTEMA), Consejo Nacional de Investigaciones Científicas y Técnicas/Universidad Nacional de Mar del Plata (CONICET/UNMdP), Av. Colón 10850, Mar del Plata B7606BWV, Argentina; tomas.byrne@intema.gob.ar (T.E.B.P.); hoppe@fi.mdp.edu.ar (C.E.H.)

2 Centre of Excellence for Nanostructured Materials (CENMAT), National Interuniversity Consortium of Materials Science and Technology (INSTM), Department of Chemical and Pharmaceutical Sciences, University of Trieste, Via L. Giorgieri 1, 34127 Trieste, Italy; diandra.mauro@phd.units.it (D.M.); daros@units.it (T.D.R.)

3 Departamento de Química, Facultad de Ingeniería, Universidad Nacional de Mar del Plata, J. B. Justo 4302, Mar del Plata B7608FDQ, Argentina

* Correspondence: julietapuig@fi.mdp.edu.ar (J.P.); faltuna@fi.mdp.edu.ar (F.I.A.)

**Abstract:** Films of a vitrimer based on the reaction between diglycidylether of bisphenol A and glutaric acid in the presence of 1-methylimidazole were processed using a solvent-based technique. The curing schedule was divided into two steps: first, a soluble linear polymer was formed through the reaction of the diacid and the diepoxide, and then the crosslinking was induced at a higher temperature via transesterification reactions. This epoxy–acid vitrimer was modified with multi-walled carbon nanotubes (MWCNTs) functionalized with β-hydroxyesters, produced by a robust and straightforward strategy based on a two-phase reaction between oxidized MWCNTs and phenylglycidylether. Nanocomposite vitrimer films were obtained by drop casting a dispersion of the functionalized MWCNTs in the linear polymer/cyclohexanone solution, followed by a thermal treatment. A high degree of dispersion of the carbon nanostructures was attained thanks to the β-hydroxyester functionalization when compared with oxidized MWCNTs. Nanocomposite films showed a significant photothermal effect (reaching 200 °C or above in 30 s) upon NIR light irradiation (850 nm) from a single LED (500 mW/cm²). The released heat was used to activate the shape memory effect and weld and heal the vitrimer matrix, proving the success of this easy strategy for the generation of remotely activated carbon-based vitrimer nanocomposites.

**Keywords:** multi-walled carbon nanotubes (MWCNTs); epoxy vitrimers; NIR photothermal effect; remote activation; shape memory; self-healing

## 1. Introduction

With a transformation driven by climate change and resource depletion, the plastic industry has begun to include sustainability considerations in the design of new materials than previously were not part of any agenda [1]. Although many of the materials used today in manufacturing industries, such as automobile or electronics industries, are recyclable or reusable, there are still countless plastic-based products that are not. This is the case of materials known as thermosets or polymeric networks that, once processed and in their final form, cannot be repaired, re-shaped, or recycled [2]. Within this group, epoxy networks play a role of major importance in the industry of coatings, paints, adhesives and high-performance composite materials with multiple applications in the aerospace, shipping, electronics, automotive, and oil industries, among many others [3–5]. One of

the oldest and more widespread uses of epoxy resins is in the design and development of coatings and paints, especially those intended to increase the durability and resistance of surfaces, improve their finishing, and prevent corrosion and environmental degradation processes. The main reasons for their popularity are their low shrinkage, high adhesion to different substrates, and the outstanding mechanical, chemical, solvent, and tribological properties typical of these crosslinked polymers [5]. However, once a conventional epoxy coating is damaged, the only way to re-establish its original performance is to reapply it. Covalent adaptable networks (CANs), due to their activated repair possibilities, are the ideal candidates to contribute to the sustainability of crosslinked coatings and prolong their service life [6]. CANs are polymeric networks that behave like a thermosetting polymer at working temperatures, but can be processed similarly to a thermoplastic one when activated by some stimuli, most commonly, heat. This is due to the dynamic nature of the crosslinking points forming the network, and it is the secret behind their reprocessability, recycling, welding ability, and self-repair.

The interest in CANs experienced an exponential growth since the seminal initial report by Leibler's group focused on the chemistry of epoxy resins cross-linked with polycarboxylic acids [7]. The reaction of an epoxy group with a carboxylic acid gives rise to an ester with a hydroxyl (OH) in the beta position. These groups can be used to produce thermally activated exchange reactions (transesterifications between an ester of one chain and the OH of another chain) in the presence of a suitable catalyst [7–9]. In these networks, called vitrimers or associative CANs, exchange reactions modify the topology of the polymer network without altering (at least theoretically) their cross-linking density. This behavior makes possible the self-healing of a part without an irreversible flow or deformation, which constitutes a significant advantage when compared with the reprocessing of a thermoplastic polymer. Hence, the advantages and special properties of CANs could be potentially transferred to a new generation of paints, coatings, and films for enhancing their durability, versatility, and functionality while diminishing their environmental impact. Recent examples of applications of CANs in coatings and films could be found in the development of self-healable paintings for the automotive industry [10], protective coatings for metal parts [11,12], flexible optical devices [13], reversible adhesives [14,15], soft actuators [16,17], and durable, ultra-thin and self-healable hydrophobic coatings [18].

The remote activation of exchange reactions in CANs can be attained through the modification of these matrices with dyes [10], metal nanostructures [19], and/or carbon fillers [11,15] thanks to the heat induced by the photothermal effect. The origin of the photothermal effect is the electronic excitation of a nanostructure or dye via irradiation at its characteristic absorption wavelength [20]. The non-radiative decay of this excitation causes the heating of the surrounding medium and activation of the repairing process, which constitutes a great advantage for the localized self-healing of intricate geometries, non-removable parts, or regions in the proximity of heat-sensitive materials [19]. In this context, it is clear that the inclusion of carbon nanostructures in CANs has the potential to enhance the properties of the latter and to add functionalities that can be used in the activation of a variety of processes, like those triggered by heat (shape memory [9,21], self-healing [19], and welding [14], among others).

As is known from the large amount of work conducted on thermosetting nanocomposites based on carbon nanostructures [22–25], the uniform dispersion of carbon nanostructures in the matrix is crucial for the best performance of these materials [26], which highlights the importance of the compatibilization/processing steps in the synthesis of the nanocomposites. Research on the synthesis and processing of nanocomposites based on vitrimers is still in its infancy, as is the understanding of the effects that nanofillers have on the recyclability, mechanical properties, and dynamic character of the network [26,27]. The use of multi-walled carbon nanotubes (MWCNTs) and other carbonaceous nanostructures is known to give improved properties to crosslinked networks, and a significant amount of effort has been invested into the design of strategies for the efficient dispersion of these materials on epoxy polymers. In the field of vitrimers, results have been scarcer. Yang et al. [15]

presented the first example of the use of MWCNTs for the photo-welding and healing of epoxy vitrimers (prepared with diglycidylether of bisphenol A (DGEBA) and adipic acid, with triazabicyclodecene (TBD) as a catalyst) triggered by the photothermal effect. The epoxy–diacid reaction has also been used for the covalent inclusion of carbon nanotubes in epoxidized natural rubber [28]. In this work, benzoic acid groups were introduced on the surface of carbon nanotubes through a diazo-coupling reaction. Despite a decrease in the stress relaxation time observed for these modified vitrimers, the ability to reprocess was demonstrated even in formulations with significant amounts of CNTs. As a different strategy to disperse CNTs in epoxy vitrimers that also conferred them enhanced electrical conductivity, polypyrrole, a conductive polymer, has been wrapped onto carbon nanotubes (CNT/PPy) and dispersed in epoxy/citric acid systems [29]. According to the results presented by the authors, these systems showed a lower effective $T_v$ than the neat ones, which was attributed to a better heat distribution in the matrix induced by the good dispersion of the CNTs. Using a similar coating approach, the interfacial interaction between MWCNTs and the epoxy matrix synthesized by reaction between DGEBA, sebacic acid, and zinc acetylacetonate as catalyst, was improved by using polydopamine, which also allowed the thermal conductivity and resistance of the epoxy vitrimer to be enhanced [30].

Although the mentioned examples have proven interesting for the production of MWCNT vitrimer nanocomposites with specific properties, it is clear that more robust, general, straightforward, and scalable procedures are still required for the production and the processing of high-performance materials based on vitrimers and carbon nanotubes. In this work, we intend to contribute to this task by presenting a new and optimized route for the fabrication of films and coatings, based on epoxy vitrimers modified with a uniform distribution of MWCNTs. We also demonstrate that the high photothermal effect of these carbon nanostructures can be efficiently used for the activation of the shape memory, self-healing, and welding of the materials under NIR radiation.

## 2. Experimental

### 2.1. Materials

An epoxy resin based on diglycidylether of bisphenol A (DGEBA; n ≈ 0.03; MW = 348.5 g/mol), glutaric acid (GA, 99%), 1-methylimidazole (1MI, 99%), phenyl-glycidylether (PGE, 98%), and triphenylphosphine (TPP, 99%) were purchased from Merck Argentina and used as received. Their chemical structures are shown in Scheme 1. The solvents cyclohexanone (CH), dimethylformamide (DMF), and tetrahydrofurane (THF) were all P.A. grade, and also acquired from Merck Argentina.

Pristine Multi-walled Carbon Nanotubes (*p*-MWCNTs), manufactured through Catalytic Chemical Vapor Deposition (CVD), with a purity of 95 wt%, an outside diameter of 20–30 nm, an inside diameter of 5–10 nm, and 10–30 μm length were purchased from Sky Spring Nanomaterials, Inc. (Houston, TX, USA) Sulfuric acid (95 wt%) and methanol were obtained from Sigma-Aldrich. Diethyl ether was from J. T. Baker (Avantor) and nitric acid at 65 wt% from Emplura (Merck Italy). Milli-Q water was produced by Millipore Milli-Q Plus 185 (Waters) purifying system. The filtration was conducted using hydrophilic PTFE filters from Millipore with a diameter size of 0.1 μm.

### 2.2. Methods

#### 2.2.1. Preparation of the Epoxy–Acid Linear Polymer and Vitrimer

The required amount of DGEBA was weighed in an aluminum mold. The epoxy monomer was heated to 100 °C under magnetic stirring (150 rpm). Then, the stoichiometric amount of GA was slowly added in small amounts to the pre-heated DGEBA. When the GA was completely dissolved (the mixture became transparent), a volume of catalyst equal to 5% in moles respect to the DGEBA equivalents was added and mixed for a few minutes under heating. The mixture was then casted in a mold and placed into an oven at 100 °C for 1 h to obtain a glassy and transparent solid material soluble in DMF and CH.

The crosslinking of this material, either after solution processing or in bulk, was attained by heating at 160 °C for 2 h.

**Scheme 1.** Chemical structures of DGEBA, GA, 1MI, TPP, and PGE.

### 2.2.2. Oxidation of *p*-MWCNTs

*p*-MWCNTs were oxidized using a previously described oxidation method with some modifications [31]. Briefly, 500 mg of *p*-MWCNTs was dispersed in a 250 mL 3:1 mixture of concentrated $H_2SO_4$ and $HNO_3$ (65%) and sonicated for 24 h at 35–40 °C. The obtained black mixture was then gradually added to 750 mL of Milli-Q water in an ice bath, followed by filtration under vacuum. Finally, the solid residue collected was washed with MeOH and $Et_2O$ and dried under vacuum. A total of 419 mg of oxidized MWCNTs (*ox*-MWCNTs) was obtained. *ox*-MWCNTs were dispersible in water, DMF, or CH.

### 2.2.3. Functionalization of *ox*-MWCNTs

β-hydroxyester-functionalized MWCNTs (*f*-MWCNTs) were obtained through an esterification reaction between *ox*-MWCNTs and PGE, following a procedure previously reported for the functionalization of silver nanoparticles (NPs) [32]. The synthetic strategy is briefly described here. PGE (270 μL) and TPP (34 mg) were added to an aqueous dispersion of *ox*-MWCNTs (50 mg in 10 mL of water), heated to 90 °C, and the emulsion maintained under vigorous magnetic stirring for 2 h. After precipitation of the *f*-MWCNTs, the supernatant was removed. Subsequently, the *f*-MWCNTs were dispersed in a minimal amount of $CHCl_3$, re-precipitated with absolute ethanol, and centrifuged to recover the solid. The dried *f*-MWCNTs were dispersible in THF, DMF, or CH.

### 2.2.4. Nanocomposite Vitrimer Film Preparation

Nanocomposites with 0.6, 1.2, 2.4, and 4.6 wt% of *f*-MWCNTs and 0.6 wt% of *ox*-MWCNTs were prepared. First, a specific amount of the linear polymer solution in CH (10 wt% or 20 wt%) was mixed with a measured volume of the *ox*-MWCNTs or *f*-MWCNTs dispersion in CH (0.5 mg/mL). To obtain the films, the dispersion was casted onto a glass slide or a Teflon-coated glass slide (to facilitate film removal), and the solvent evaporated at 65 °C. After film formation, a two-hour treatment at 160 °C was carried out to induce crosslinking by transesterification.

*2.3. Characterization Techniques*

2.3.1. Fourier-Transformed Infrared (FT-IR)

A Nicolet 6700 FT-IR spectrometer was used for acquiring FT-MIR spectra (4000 to 400 cm$^{-1}$ range) of the epoxy–acid vitrimer after 1 h of reaction at 100 °C. A drop of the solution in DMF was casted on a KBr window and dried before acquiring the spectra. This procedure was also employed to characterize the organic groups present on the surface of MWCNTs. Evolution of the epoxy conversion was also followed in the near-infrared region (FT-NIR, 8000 to 4000 cm$^{-1}$, transmission mode). In this case, samples with a thickness of 1.5–2 mm were analyzed. All spectra were acquired with a resolution of 4 cm$^{-1}$.

2.3.2. Dynamic–Mechanical Analysis (DMA) and Stress Relaxation

Dynamic–mechanical analysis (DMA) was carried out using an Anton Paar Physica MCR301 rheometer under a torsion fixture. Rectangular cross-section (1.5 mm × 6 mm) specimens with a span of 18 mm were tested. An oscillation frequency of 1 Hz and a deformation amplitude of 0.05% were used, from −60 °C to 150 °C at a heating rate of 5 °C/min. The same configuration and specimen dimensions were used for the stress relaxation test at 160 °C, with a deformation of 5%.

2.3.3. Thermogravimetric Analysis (TGA)

Thermogravimetric analysis (TGA) was carried out to determine the organic fraction in the MWCNTs. TGA thermograms were acquired using a Shimadzu TGA-50 at a heating rate of 10 °C/min under nitrogen flow up to 600 °C.

2.3.4. Transmission Electronic Microscopy (TEM)

Transmission Electron Microscopy (TEM) images were obtained using an EM 208 microscope (Philips, United Kingdom) equipped with Quemesa camera (Olympus Soft Imaging Solutions). The images were acquired at a voltage of 100 kV using RADIUS 2.1 software. The samples were prepared by drop-casting an ethanol suspension of MWCNTs on the TEM grids (LC300-CU, Lacey Carbon Film, Copper, 300 mesh) followed by drying under vacuum overnight. To prevent agglomeration of MWCNTs, the solution was sonicated for 30 min before deposition.

2.3.5. Dynamic Scanning Calorimetry (DSC)

DSC analyses were carried out on a Perkin Elmer Pyris 1 under a constant flow of N$_2$. A first scan was performed for each sample from 10 °C to 160 °C, at a heating rate of 10 °C/min, in order to erase the thermal history and to eliminate the moisture absorbed during the storage between the synthesis and the testing of the materials. The glass transition temperature, Tg, of each material was measured at the onset from a second temperature scan from 20 °C to 100 °C, at a heating rate of 10 °C/min.

2.3.6. Transmission Optical Microscopy (TOM)

A Leica DMLB transmission optical microscope provided with a video camera Leica DC100 (Wetzlar, Germany) was used to analyze the uniformity of the dispersion of MWCNTs into the polymer matrix at a micrometric level. The same arrangement was used for observing the superficial and deep scratches, made with the tip of a needle or a cutter, respectively, on the film surface, and for their recovery after heating via irradiation.

2.3.7. Profilometry Measurements

A KLA Tencor D-100 Profilometer was used to scan the topographic profile of superficial and deep scratches on the damaged film and its evolution after irradiation.

2.3.8. Photothermal Effect Measurement

The heating response of the samples synthesized with different contents of *f*-MWCNTs was characterized via irradiation, at a variety of times times, with a LED source (λ = 850 nm,

500 mW/cm$^2$). The thermal response was recorded with a Perfect Prime IR0019 handheld thermal camera provided with a 320 × 240 IR resolution and a frame rate of 9 Hz.

### 2.3.9. Raman Analysis of MWCNTs

Samples of *p*-MWCNTs and *ox*-MWCNTs were dispersed in absolute EtOH and *f*-MWCNTs in CH. After sonication, dispersions were drop-casted on Si/SiO$_2$ supports (Thermal Oxide Silicon Wafers, Nanovision, 100 mm diameter, oxide thickness 300 nm, wafer thickness 500 μm, resistivity 0.001–0.005 ohm-cm, type/dopant P/boron) and left to dry. The silicon wafers were placed on microscope slides (VWR 72 Polysine, Cat. No. 631-0107). Raman spectra were collected using a Renishaw in Via Raman Microscope with a 532 nm green laser (laser power 10, exposure time 10 s, 20 × objective and number of acquisitions 5).

### 2.3.10. Shape Memory and Welding Experiments

Rectangular samples of self-standing films (~100 μm in thickness) containing 1.2 wt% of *f*-MWCNTs were heated above their Tg via irradiation for 30 s using an IR lamp (France) with an output power of 150 W placed 10–12 cm from the sample, and immediately twisted into a new shape by imposing a small stress by hand. Cooling of the sample at room temperature under the imposed stress produced a fixed secondary shape. Recovery of the initial shape by irradiation was recorded on video. Welding of two films, superimposed forming a cross, was qualitatively assessed by analyzing the resistance of the joint, formed after irradiation for 10 min with an IR lamp.

## 3. Results and Discussion

### 3.1. Synthesis and Characterization of the Vitrimers

As previously discussed in the introduction, self-healable remotely activated coatings and films with the potential to be applied from a solution via typical processing techniques like dip coating, spraying, or spin coating are desirable materials for the fabrication and scaling-up of a variety of devices. The strategy proposed in this work for producing this type of film is based on the use of a non-common curing and processing protocol of a well-known epoxy–acid chemistry. The developed procedure enabled the easy modification and processing of these self-healing and reprocessable materials with multi-walled carbon nanotubes (MWCNTs), functionalized via a robust and scalable technique, to form NIR-activated vitrimer nanocomposite films. The reaction between a difunctional epoxy monomer, diglycidylether of bisphenol A (DGEBA), with a dicarboxylic acid, glutaric acid, GA, in presence of 1-methylimidazole (1MI) as catalyst, produces a linear polymer (Scheme 2a) through an epoxy–acid polyaddition [9]. This is the expected result of the reaction between two difunctional monomers for short times at moderate temperatures in absence of dynamic bond exchanges [5,33,34] and it is, in fact, what is obtained when these two reagents are heated in bulk for one hour at 100 °C. The FT-MIR spectrum taken after 1 h of reaction at 100 °C (Figure 1) shows the consumption of both characteristic peaks corresponding to the epoxy antisymmetric deformation (ν = 915 cm$^{-1}$) and carboxylic acid antisymmetric stretching (ν = 1710 cm$^{-1}$) groups and the formation of the ester with a characteristic band at 1734 cm$^{-1}$ (insets). A wide and intense band corresponding to the formation of hydroxyl groups is clearly observed at 3400 cm$^{-1}$. The complete conversion of the epoxy group could also be verified using FT-NIR (Figure 2) following the characteristic epoxy band centered at ν = 4532 cm$^{-1}$ (corresponding to the conjugated epoxy CH$_2$ deformation band with the aromatic CH fundamental stretch). The epoxy–acid reaction can be considered complete at this point, with complete solubility of the material in DMF or CH and no significant gel fraction detected.

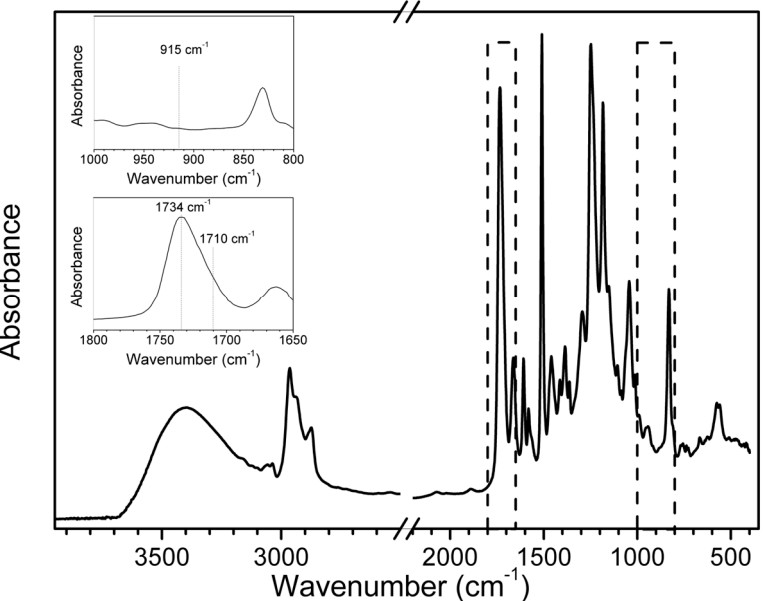

**Scheme 2.** (**a**) Formation of the linear poly β-hydroxyester through the epoxy–acid addition catalyzed by 1MI; (**b**) transesterification reactions leading to the formation of covalent crosslinks.

**Figure 1.** FT-MIR spectrum of the polymer obtained after 1 h of bulk polymerization at 100 °C. Insets depict selected regions (marked with a dashed square) showing the bands associated with epoxy (915 cm$^{-1}$), carboxylic acid (1710 cm$^{-1}$), and ester groups (1734 cm$^{-1}$).

The network formation occurs in a second step via transesterification reactions between the β-hydroxyesters formed in the first step, which produce crosslinks between chains. This process occurs when transesterification reactions are activated via heating of the polymeric material at higher temperatures for longer times (Scheme 2b), and is allowed by the dynamic nature of the β-hydroxyester bonds in the presence of 1MI. The bond exchanges lead to the rearrangement of the linear chains into a network, also leaving pendant chains and a small soluble fraction. For a deeper statistical analysis of the gel formation and crosslinking through exchange reactions, the reader is directed to a previous article from our research group [34]. In the case of the polymer herein described, the crosslinking and formation of the matrix can be verified after an additional thermal treatment of 2 h at 160 °C. After this treatment, the material can no longer be completely dissolved, and the FTIR spectrum shows essentially no variation. These results were confirmed using DMA measurements. The storage and loss moduli and the damping factor corresponding to materials before and after the second step of 2 h at 160 °C are shown in Figure 3.

The thermoplastic polymer obtained at 100 °C shows a sharp drop in the storage modulus starting at 45 °C, and the associated peak in the damping factor, corresponding to the alpha transition temperature (Tα), is centered at 58 °C. After the transition, the material starts to flow and the G′ decreases to values below the measurement threshold of the device. This is the typical behavior of a non-crosslinked polymer (a thermoplastic behavior) that, upon heating above its glass transition/melting temperature, becomes a liquid, which is also evident in the loss modulus being higher than the storage modulus (or tan δ > 1) at temperatures above 70 °C. After heating at 160 °C for 2 h, the response of the polymer changes, showing the characteristic rubbery plateau of crosslinked polymers. The Tα value remains essentially unchanged, with the slight difference of about 1 °C ascribed to the moisture absorption observed for these materials when stored for several days.

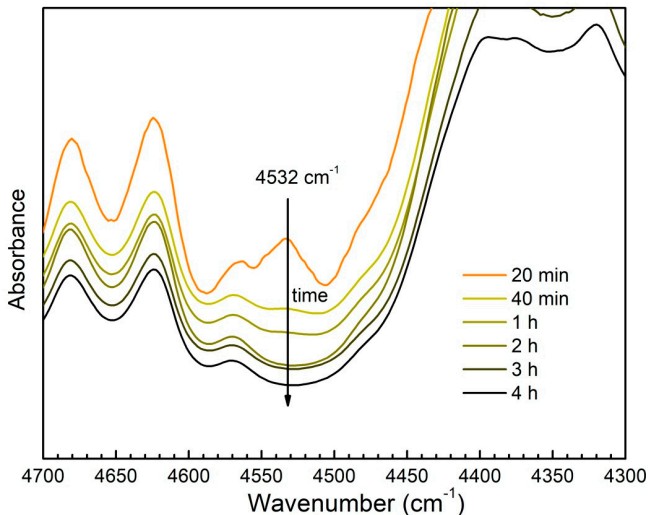

**Figure 2.** Evolution of the epoxy conversion followed by the decrease in the epoxy NIR peak centered at 4532 cm$^{-1}$.

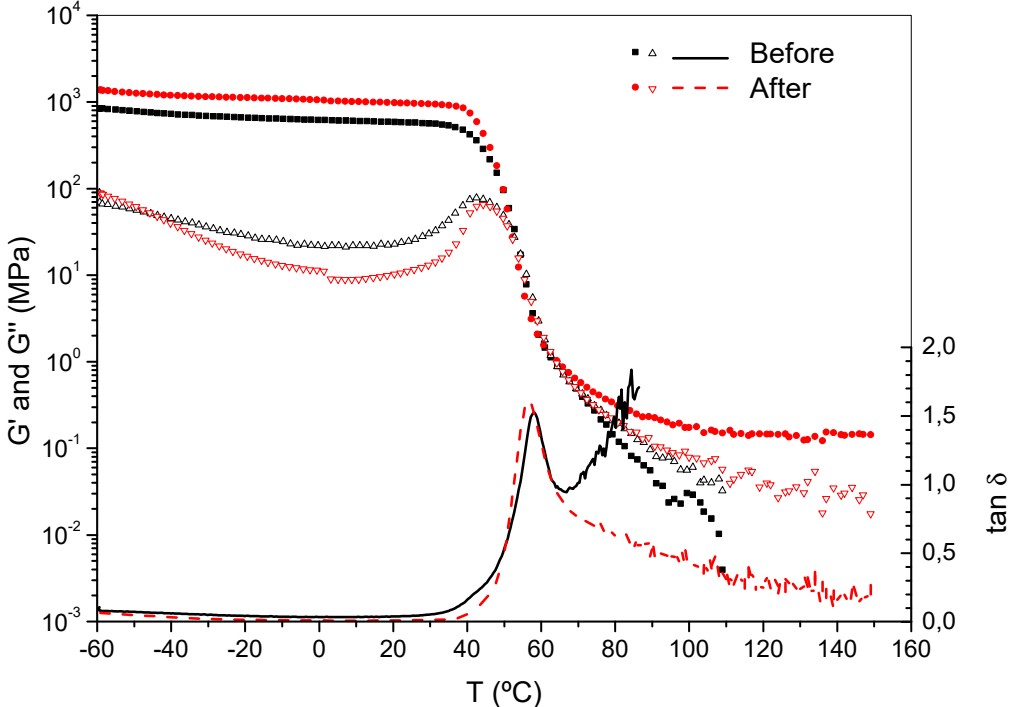

**Figure 3.** Storage (G′; full symbols) and loss (G″; empty symbols) shear moduli and damping factor (tan δ; dashed and full lines) for DGEBA-GA-1MI polymer before and after the step at 160 °C for 2 h.

Figure 4 shows the stress relaxation at 160 °C of the neat DGEBA-GA-1MI vitrimer after the complete curing cycle. The system is able to quickly relax, with a characteristic time of 8 min 10 s, which is significantly lower than the corresponding values for other epoxy–acid vitrimers using $Zn^{+2}$ [35,36], TBD [37] or tertiary amines [8,38] as catalysts at similar temperatures.

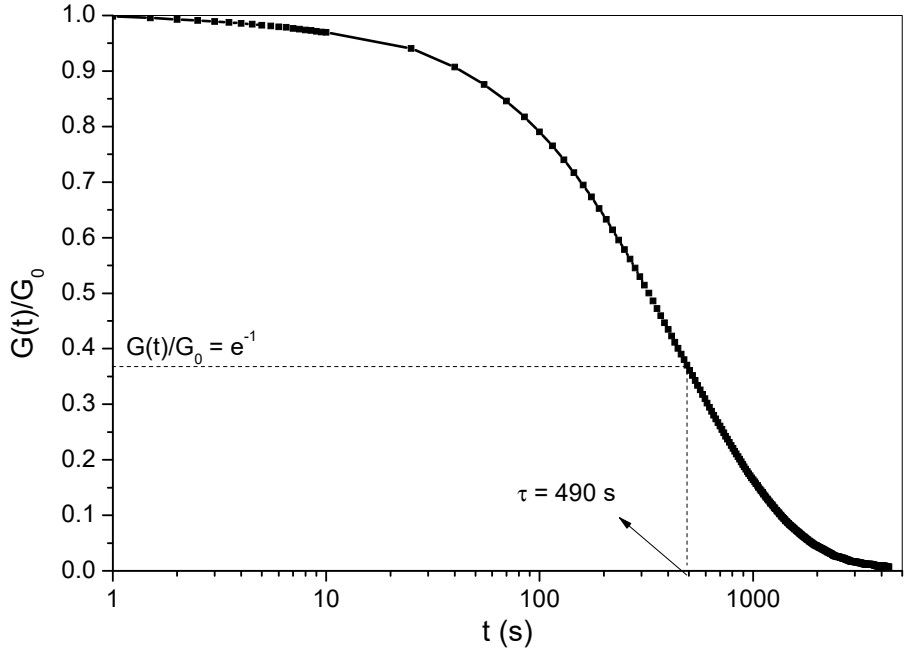

**Figure 4.** Stress relaxation of the DGEBA-GA-1MI network at 160 °C.

### 3.2. Films' Processing

The formation of a linear polymer in the first curing step allows the total dissolution of the material in solvents, as it is expected for any thermoplastic polymer. After a preliminary screening, DMF and CH were selected as the best solvents for this material, with a fast dissolution rate and a good film-forming ability. Films were fabricated using drop casting, although the dip-coating technique was also tested in these systems (Figure 5) to demonstrate the potential use of more scalable techniques, which are useful in industrial applications (the scale-up and optimization of these formulations for spray coating, spin coating, and dip coating is currently underway and will be the subject of a future publication). The complete elimination of the solvent was carried out via evaporation overnight at 65 °C. After this step, the film is still a thermoplastic material that can be transformed into a crosslinked polymer by heating at 160 °C for 2 h.

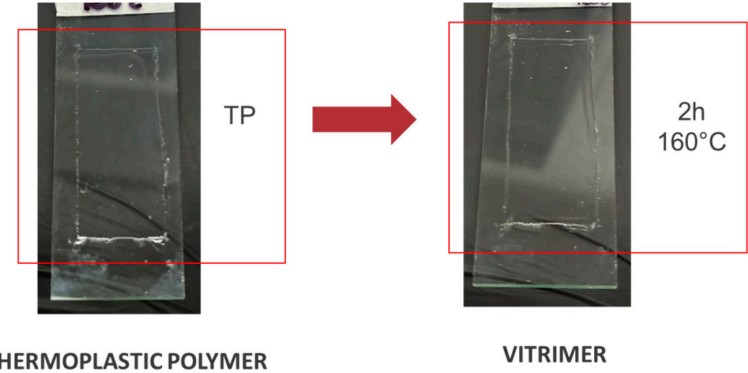

**THERMOPLASTIC POLYMER**　　　　　　**VITRIMER**

**Figure 5.** Optical images of the vitrimer films prepared by dip coating from 10 wt% solutions in DMF of the polymer obtained after 1 h of reaction at 100 °C.

### 3.3. Functionalization of MWCNTs

Although MWCNTs show a highly efficient photothermal effect, the efficient transfer of this property to a polymeric matrix is not a trivial task, mainly because their dispersion is usually poor due to their strong attractive van der Waals forces and large aspect ratios, which cause them to agglomerate [26,30]. As also discussed in the introduction, the MWCNTs' surface can be chemically modified to improve the dispersion within a polymer or vitrimer matrix [28,39–41]. Considering only the chemical affinity between filler and matrix, it could be argued that the use of *ox*-MWCNTs, containing –COOH and –OH groups, that can covalently bond to the matrix through transesterification reactions, could be enough for giving a nice dispersion. With this idea, oxidation was performed on *p*-MWCNTs. Chemical oxidation in an acid medium is a typical method used to modify the surface of MWCNTs, where oxygenated functional groups (such –COOH, –C=O, and –OH) are introduced [41,42]. Figure 6a shows optical images of the aqueous dispersions of *p*-MWCNTs (left) and *ox*-MWCNTs (right). A marked change in colloidal stability in water is clearly observed for *ox*-MWCNTs with respect to *p*-MWCNTs, which can be attributed to the high increase in hydrophilicity of oxidized nanotubes, possibly linked to the presence of –COOH and –OH groups on the surface. Figure 6b shows transmission electron microscopy (TEM) micrographs of *p*-MWCNTs and *ox*-MWCNTs. A shortening in the length of these nanostructures is clearly observed after oxidation, which is a common effect for this type of procedure when oxidation takes place in strong conditions [43–46].

Initial tests carried out by blending *ox*-MWCNTs in DMF or CH solutions of the polymer (or even in the melt), proved that the proposed approach was unviable (Figure 7). The simple oxidation of MWCNTs was not enough to achieve a good dispersion in the polymer matrix, neither in the initial formulation nor in the obtained linear epoxy/acid polymer processed from the solution. Figure 7 shows the presence of macroscopic aggregates (even observable by the naked eye) of the fillers in the polymer.

After demonstrating that oxidation was not a convenient route for the dispersion of these MWCNTs in the matrix, a different strategy was proposed, based on the formation of a β-hydroxyester group on the surface of the MWCNTs, with the idea of preparing a functionalization with a molecular structure more similar to that of the polymer network. The esterification reaction takes place between the carboxylic acid group of the *ox*-MWCNTs and the epoxy ring of the phenyl glycidyl ether (PGE) to produce a β-hydroxyester group with an aromatic ring (Scheme 3) [32] that can also be bond to the network through transesterification reactions taking place in the second step performed at 160 °C.

The esterification reaction occurred in two phases, with the aqueous phase being the dispersion of *ox*-MWCNTs and the organic phase the blend between PGE and TPP. The reaction proved to be very convenient. No organic solvents or prolonged heating at high temperatures were required for esterification (as previously reported for other functionalization protocols) [42]. After only 30 min of the start of the reaction, the precipitation of a black solid was clearly observed, pointing to a change in hydrophilicity of the colloid from a more hydrophilic (*ox*-MWCNTs) to a less hydrophilic behavior (*f*-MWCNTs). After 2 h of heating at 90 °C, the aspect of the dispersion did not show further changes. The almost clear aqueous supernatant was removed and, after centrifugation and washing, the *f*-MWCNTs could be easily dispersed in organic solvents such as chloroform, DMF, and CH. Water dispersibility was lost at this point.

The oxidation process and the esterification reaction of MWCNTs were analyzed using FTIR and TGA. Figure 8 shows the FTIR spectra of *ox*-MWCNTs and *f*-MWCNTs, showing a characteristic peak at 1580 cm$^{-1}$ in both types of MWCNTs, which corresponds to the C=C stretching band [42]. For *ox*-MWCNTs, the band at 1715 cm$^{-1}$, associated with the C=O stretching of the carboxylic acid group, is clearly visible but disappears for *f*-MWCNTs after the chemical functionalization, being replaced by a new band at 1734 cm$^{-1}$ assigned to the C=O stretching of the ester group and confirming the esterification reaction [42,47]. The band at 1107 cm$^{-1}$, present in the *ox*-MWCNTs and corresponding to the C−O stretching

vibrations of the carboxylic acid [48,49], also disappeared in *f*-MWCNTs, proving the consumption of these groups in the esterified nanostructures.

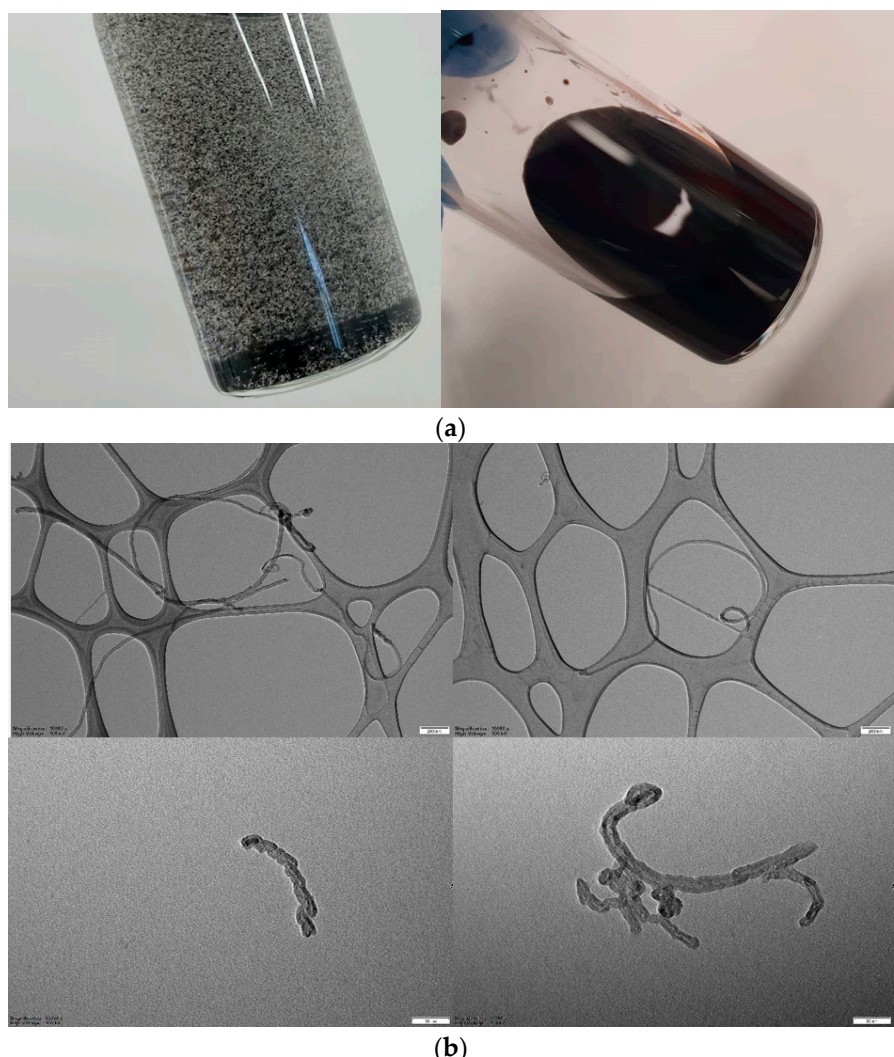

**Figure 6.** (**a**) Optical images of aqueous dispersions of *p*-MWCNTs (**left**) and *ox*-MWCNTs (**right**). (**b**) TEM images of *p*-MWCNTs ((**top**), scale bar = 200 nm) and *ox*-MWCNTs ((**bottom**), scale bar = 50 nm).

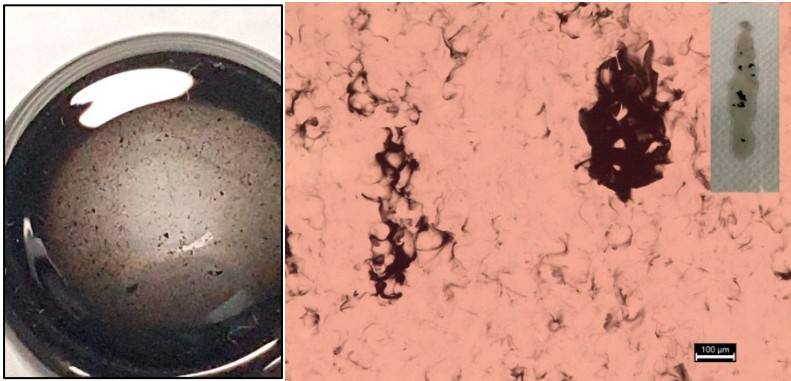

**Figure 7.** Optical image of *ox*-MWCNTs (0.5 wt%) in the unreacted epoxy/acid formulation (**left**). Optical micrograph of *ox*-MWCNTs (0.5 wt%) in the films obtained from solutions of the thermoplastic polymer (inset: optical image) (**right**). The scale bar is 100 μm.

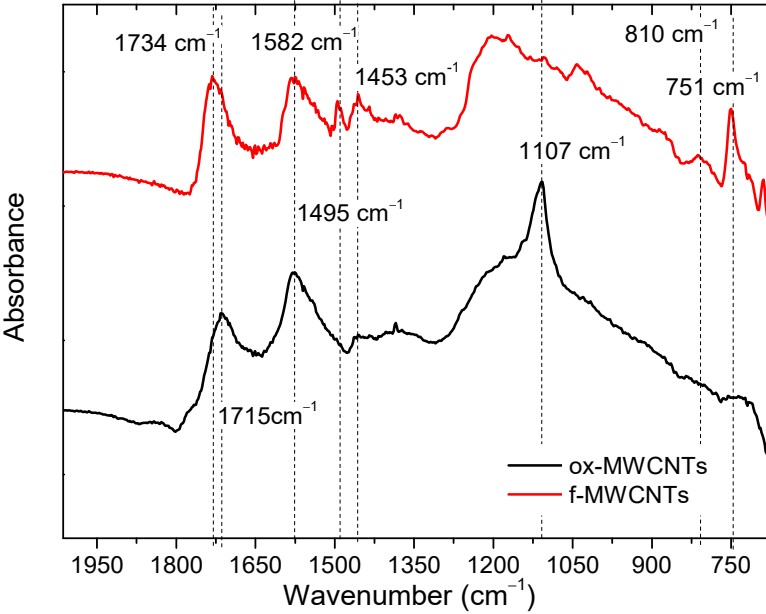

**Scheme 3.** Reaction of a carboxylic acid group with the epoxy group of the PGE leading to a β-hydroxyester functional group.

**Figure 8.** FTIR spectra of oxidized and functionalized MWCNTs.

The presence of the product of the esterification (β-hydroxyester) was also demonstrated by the assignments at 810 cm$^{-1}$, which corresponds to the C-O skeletal stretching band of a secondary alcohol, the band at 751 cm$^{-1}$ that corresponds to C-H wagging vibrations of the phenyl group, and the bands at 1453 cm$^{-1}$ (partially), and 1495 cm$^{-1}$, which correspond to the bending modes of CH$_2$ groups [41,50].

According to the thermograms of the MWCNTs in Figure 9, the mass fraction of organic groups removed (measured at 450 °C) for *ox*-MWCNTs is 10% and for *f*-MWCNTS is 20.3%. For the sake of comparison, Chen et al. [41] reported a slightly lower content of –COOH and a moderately higher amount of organic fraction (around 28%) after esterification in DMF for 36 h. Under the assumption that the entire organic fraction arises from –COOH functionalities, the corresponding degree of functionalization of the *ox*-MWCNTs is 2 mmol/g in carboxylic functions. The complete esterification of these groups should give around 32.5% of organic content for the *f*-MWCNTs. The obtained value, 20.3%, would indicate that the esterification efficiency is about 40%. However, it is important to note that this value could be higher if the initial organic content of the ox-MWCNTs is not only constituted by –COOH groups. In summary, the obtained values show that esterification occurs to a significant extent on the surface of the carbon nanotubes.

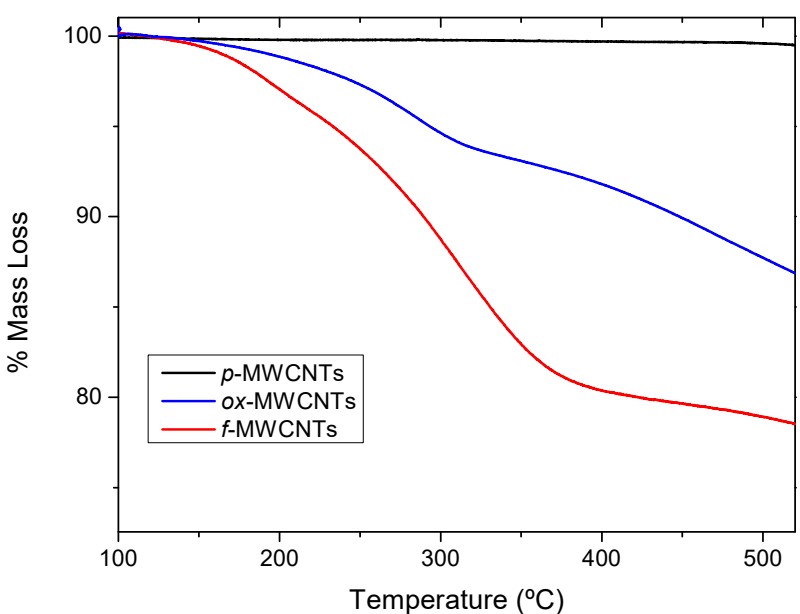

**Figure 9.** TGA thermograms of pristine, oxidized, and functionalized MWCNTs.

Raman spectra were also used to characterize the MWCNTs. The Raman spectra of *p*-MWCNTs, *ox*-MWCNTs, and *f*-MWCNTs, shown in Figure 10, exhibit two peaks at around 1350 cm$^{-1}$ and 1582 cm$^{-1}$ corresponding to the defined D band and G band, respectively. From Figure 10, it can be seen that the D band revealed a clear increase in intensity after the oxidation and esterification processes. A larger ratio between the D and G bands ($I_D/I_G$), ranging from 0.84 to 1.11 to 1.21, (from *p*-MWCNTs to *ox*-MWCNTs to *f*-MWCNTs, respectively) is usually associated with a higher degree of disorder and defects in the structure, indicating a greater degree of covalent functionalization in MWCNTs [51].

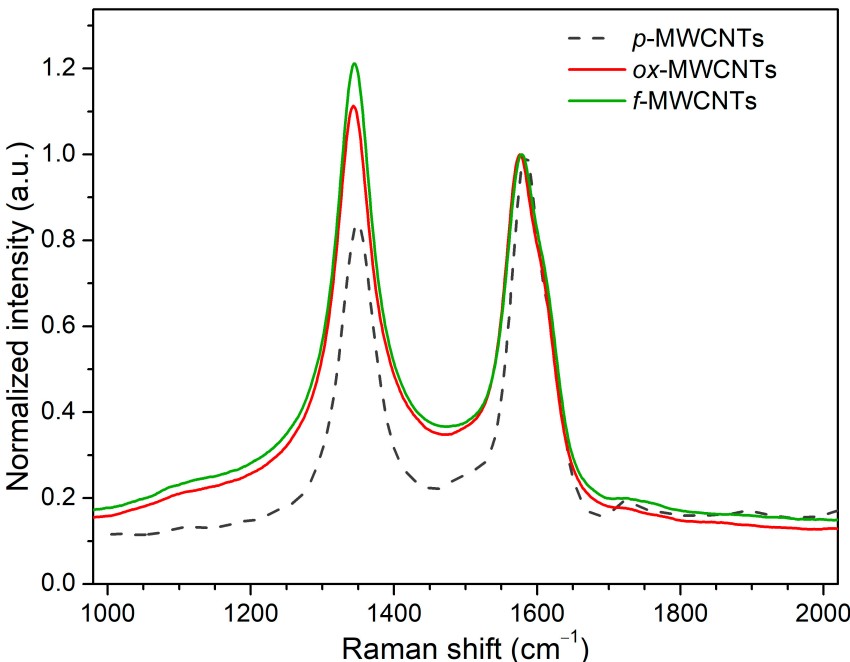

**Figure 10.** Raman spectra and $I_D/I_G$ values for *p*-MWCNTs, *ox*-MWCNTs, and *f*-MWCNTs.

As a summary of this part, the results obtained after functionalization showed that esterification of the *ox*-MWCNTs was successful and took place through a very convenient,

easy and fast protocol. The use of these *f*-MWCNTs as modifiers of vitrimers will be discussed in the following sections.

### 3.4. Dispersion of MWCNTs in the Vitrimer

Selected amounts of *f*-MWCNTs were dispersed in 10 wt% or 20 wt% solutions of the polymer in CH, as described in the experimental section. After the evaporation of the solvent, self-standing films and coatings of the modified vitrimers were obtained. Figure 11 presents optical micrographs showing a comparison between the films containing 1.2 wt% of *ox*-MWCNTs and *f*-MWCNTs, respectively. As previously shown, aggregates of more than 100 μm in size were formed in samples prepared with *ox*-MWCNTs, whereas a much better dispersion was attained with the same content of *f*-MWCNTs. These results evidenced the existence of chemical compatibilization mechanisms operating in these samples. The structural similarities between the β-hydroxyester moieties, generated on the surface of the MWCNTs, and the network polymer segments are the first possible reasons for the improved dispersion of the nanotubes in the matrix. On the other hand, the possibility of the formation of covalent bonds between these *f*-MWCNTs and the network through exchange reactions (transesterifications between hydroxyesters), during the heating step a 160 °C for 2 h cannot be ruled out. This covalent bonding has also been postulated as responsible for the homogeneous dispersion of MWCNTs in other epoxy matrices [41,50]. The successful incorporation and strong bonding of *f*-MWCNTs into the epoxy–acid matrix, as well as the crosslinking state of the polymer after thermal treatment at 160 °C, is also evidenced by the aspect of the nanocomposite immersed in DMF. Storage for prolonged time in this excellent solvent showed no evidences of nanotubes migration or dissolution of the matrix, pointing to a strong physical and/or chemical affinity between the MWCNTs and the network (Figure 11c).

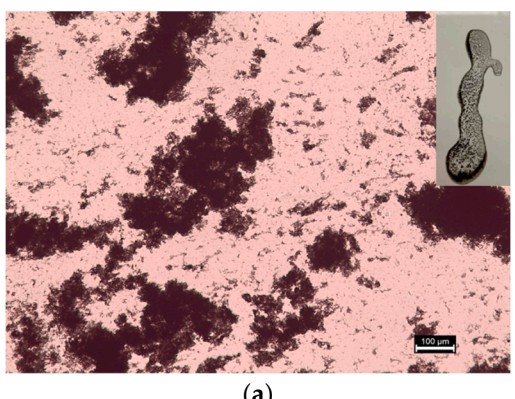 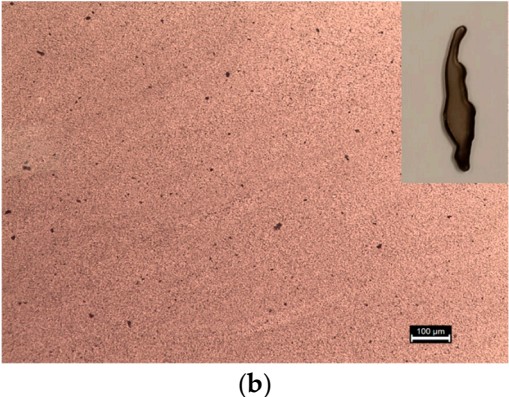 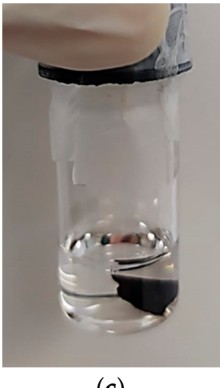

(**a**)                                               (**b**)                                               (**c**)

**Figure 11.** TOM images of nanocomposite vitrimers with 1.2 wt% of *ox*-MWCNTs (**a**) and 1.2 wt% of *f*-MWCNTs (**b**) casted onto glass slides. The bar is 100 μm. Insets show an optical image of the coatings produced by the irregular dropping of the solution on glass slides, followed by evaporation and thermal treatment at 160 °C. (**c**) Optical image of the nanocomposite immersed in DMF for a prolonged time.

The influence of the incorporation of *f*-MWCNTs on the thermal properties of nanocomposite vitrimers was analyzed via DSC (Figure 12). The glass transition temperature (Tg) of the modified samples increased from 44.5 °C (neat epoxy matrix) to 48.8 °C, 49.5 °C, and 48.0 °C with the incorporation of 1.2, 2.4, and 4.6 wt% of *f*-MWCNTs, respectively. The increase in Tg observed for higher concentrations of fillers would indicate that *f*-MWCNTs could be forming part of additional chemical or physical crosslinking points that restrict the movement of the polymer segments. As previously discussed, crosslinking points could be formed by a chemical reaction between *f*-MWCNTs or physical, strong interaction between these fillers and the matrix. Further investigations would be necessary to elucidate the nature of these not-easy-to-track interactions, although this is outside the aim of this work.

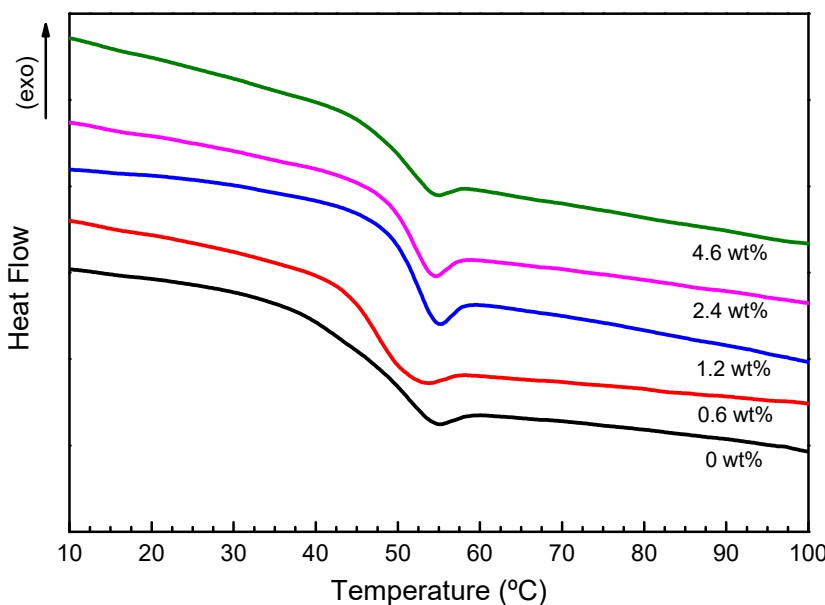

**Figure 12.** DSC thermograms of nanocomposites with different contents of *f*-MWCNTs.

### 3.5. Photothermal Effect: NIR Activation of Welding, Self-Healing, and Shape Memory

The ability of the nanocomposite vitrimer films to be indirectly heated through the photothermal effect was determined by carrying out thermal measurements upon irradiation with an infrared LED at different times. Figure 13 shows the measured temperature change as a function of time for nanocomposites with *f*-MWCNTs contents ranging from 0.6 to 4.6 wt.%, and for the neat vitrimer as a control experiment. As it can be seen, under an estimated irradiance of 500 mW/cm$^2$ and a wavelength of 850 nm, the temperature rises very quickly, reaching values of 90–170 °C (depending on the *f*-MWCNTs content) in only 5 s, and attaining a steady state at temperatures between 160 °C and 200 °C in less than half a minute. The neat vitrimer also showed a small temperature increase (about 25 °C), probably due to the heat produced by the LED device placed a few millimeters away from the samples. These results show that it is possible to produce nanocomposites with a high photothermal response through the uniform dispersion of *f*-MWCNTs in epoxy–acid vitrimers. This temperature increase can be regulated by tuning the irradiance, distance of the irradiation source from the sample, and filler content to produce materials with variable performances. The use of "safe", penetrating NIR light enhances the range of applications of these systems and makes these materials potential candidates for high-performance devices. As a first example of this potentiality, the shape memory effect of these networks was activated by irradiation with an infrared light bulb lamp to remotely induce a change in shape (Figure 14a; Video S1). A rectangular film of the nanocomposite containing 1.2 wt% of *f*-MWCNTs was irradiated for 5 sec, allowing it to reach a temperature above its Tg. At this point the material was folded and twisted to obtain a secondary shape that was fixed by cooling it at room temperature by imposing a minimal tension with the hands. At this glassy state, the material stores elastic energy in chain conformations that are different from those at the equilibrium [52]. After the fixation of this secondary shape, irradiation was used for the recovery of the original shape, because chains returned to the conformations of equilibrium. The recuperation of the original shape was attained in 2–3 s.

Due to the dynamic nature of the vitrimeric matrix and to the presence of MWCNTs, remotely activated heating has the potentiality of being used for the welding and self-healing of parts via simple irradiation with NIR light. In a proof of concept experiment, two pieces of the nanocomposite were bonded together in a localized region (marked with a dashed circle) via irradiation with an IR LED for 10 min (Figure 14b). The strength of the welding was tested by pulling from both of the welded pieces at the same time, leading to a cohesive failure, without any debonding observed (Figure 14b and Video S2). This

experiment proved the high efficiency of the welding, which takes place even in the absence of an external force that keeps a good contact between the pieces.

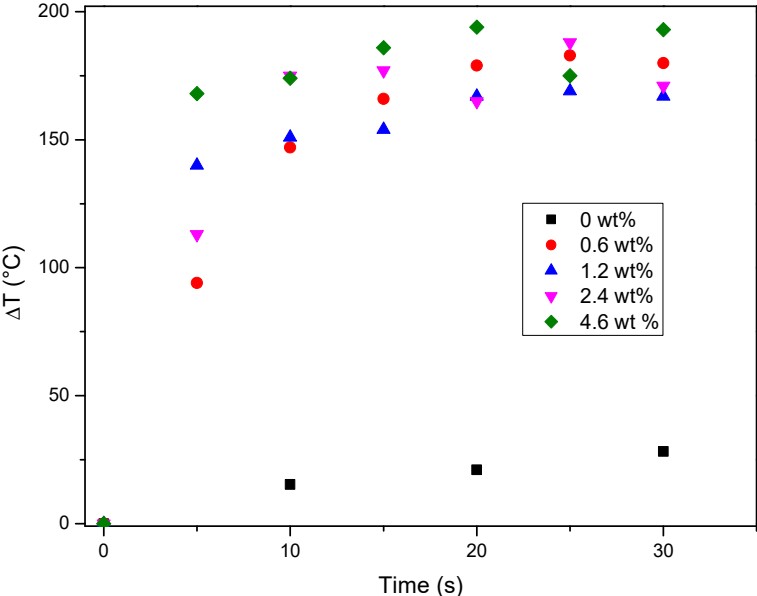

**Figure 13.** Temperature increase of the samples with different wt% of *f*-MWCNTs irradiated with a LED lamp in the NIR region ($\lambda$ = 850 nm, 500 mW/cm$^2$).

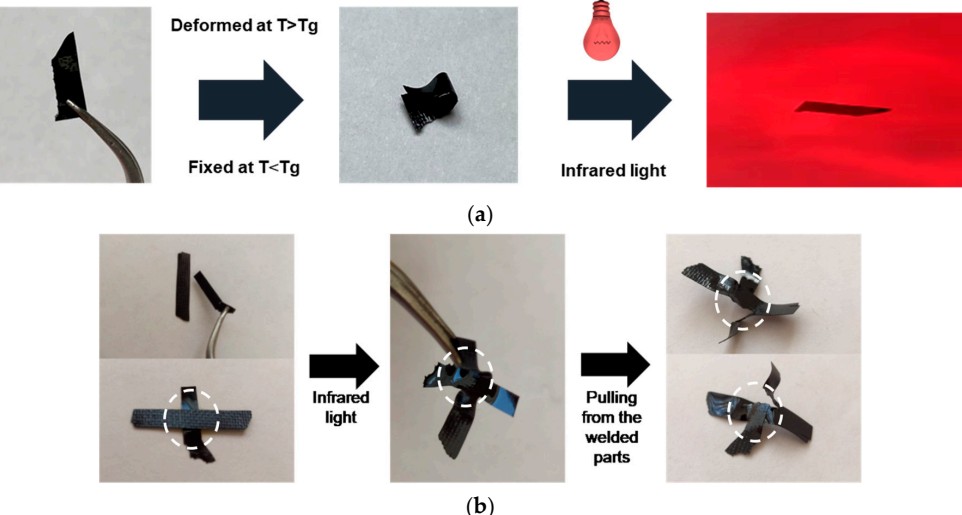

**Figure 14.** Optical images showing the NIR-activated shape memory effect (**a**) and welding of two films via irradiation (**b**).

Finally, the IR LED irradiation was also used to repair cracks of different severity performed on a vitrimer nanocomposite coating deposited on a glass slide, containing 1.2 wt% of *f*-MWCNTs. Superficial scratches were made by gently sliding a hypodermic needle on the nanocomposite surface while exerting a slight force. The superficial scratches completely disappeared after only 10 min of irradiation, as could be observed through TOM and in profilometry measurements (Figure 15). Deeper and harsher cuts were inflicted on the film using a cutter and applying a higher force, removing material to expose the glass substrate. In this case, though the TOM observations and profilometry scans clearly showed that the crack depth diminished and the damage is healed to an acceptable extent after only 10 min of irradiation (Figure 16), further treatment did not show an improvement on the reparation. This is likely due to the crack being too wide when compared to the film thickness, which prevents the damaged surfaces from coming into contact with each other.

A small amount of acetone (approximately 50 μL) deposited on the crack aided to a closer contact thanks to the swelling of the vitrimer matrix [53], producing a better healing after evaporation and subsequent heating at 160 °C for 1 h (Figure S1).

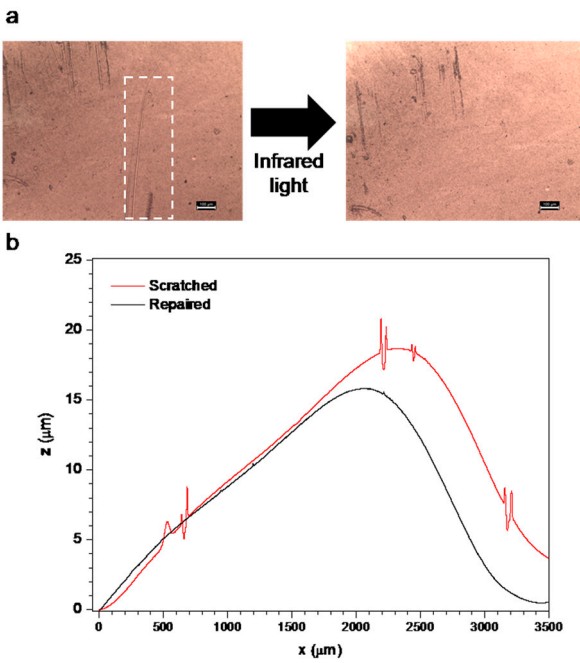

**Figure 15.** (**a**) TOM images (scale bar = 100 μm) and (**b**) profilometry scans of the film damaged with a needle and healed after 10 min of irradiation with the infrared LED.

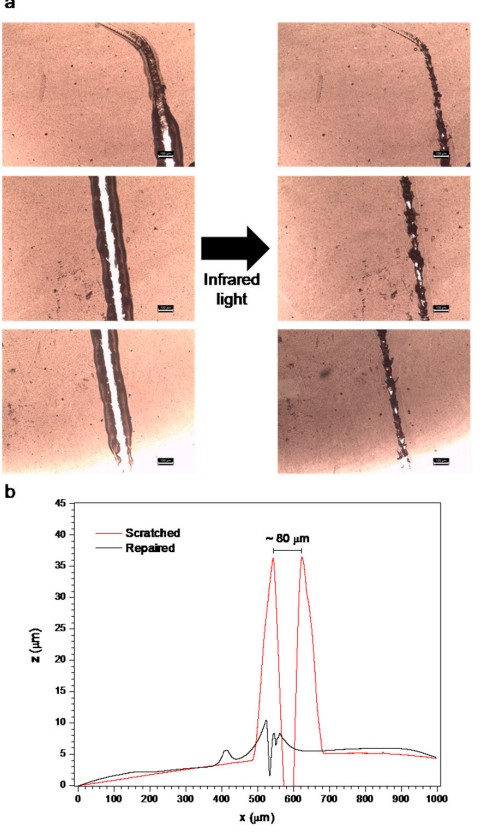

**Figure 16.** (**a**) TOM images (scale bar = 100 μm) and (**b**) profilometry scans of the film damaged with a cutter and healed after 10 min of irradiation with the infrared LED.

Finally, the possibility of reprocessing the nanocomposite via thermal treatment was analyzed to evaluate the possibility of recycling the fabricated self-standing films. With this aim, four different pieces cut from the material previously broken in the LED welding test were placed together between two Teflon-covered glass plaques, secured with clamps, and heated in a convection oven for 2 h at 160 °C. Figure 17 shows the four pieces bonded into a single one after the heating, with a surface pattern copied from that of the Teflon coating of the glass plaques.

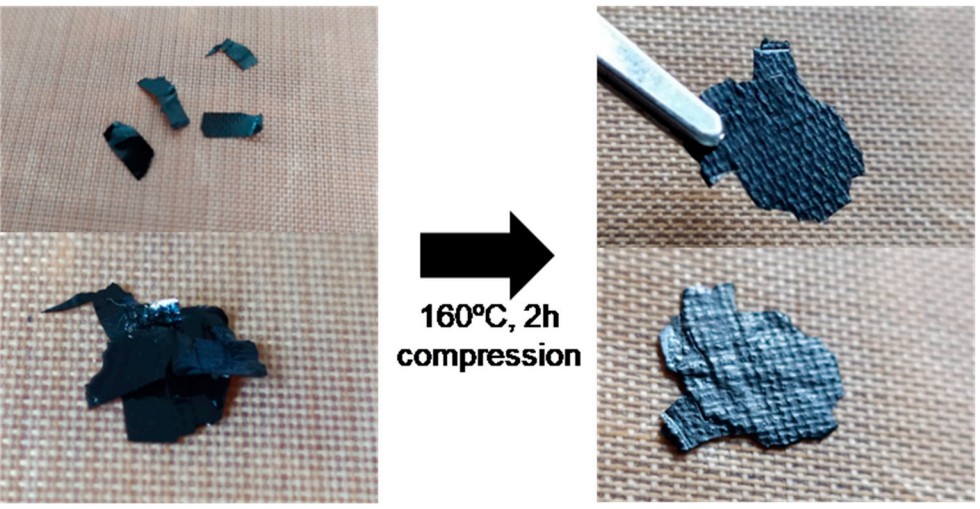

**Figure 17.** Reprocessing of the DGEBA-GA-1MI/$f$-MWCNTs nanocomposites for 2 h at 160 °C.

## 4. Conclusions

A robust, straightforward, and convenient strategy for the preparation, processing, and modification of vitrimer nanocomposite films was developed. The success of the approach was based on two main points involving aspects of the synthesis of the vitrimer and the MWCNTs functionalization. First, the design of a proper route for the generation of the vitrimer films, based on the separation of the curing schedule in two differentiated steps, enabled the synthesis of a linear polymer that acted as precursor of the vitrimer networks. This polymer can be processed in solution, in the same way in which thermoplastic polymers are usually processed to produce films and coatings. After this step, transesterification activated by a subsequent heating treatment at 160 °C produced a dynamic network. The second important aspect of the proposed strategy is the transformation of these polymeric precursors into NIR-activated nanocomposites by the addition of adequately functionalized MWCNTs. A high degree of dispersion and an optimal photothermal response could be attained by following a very easy and fast functionalization procedure based on an esterification reaction performed in two phases. The convenience of this procedure means that it could be extended to the functionalization of other carbon-based nanostructures that require dispersion in similar matrices. Finally, the ability of the systems to be thermally reprocessed and remotely self-healed, welded, and actuated by using NIR light is demonstrated, opening the door to the development of new adhesives, protective coatings, soft actuators, and other smart and high-performance devices for multiple technological applications.

**Supplementary Materials:** The following supporting information can be downloaded at: https://www.mdpi.com/article/10.3390/c9040119/s1, Video S1. Shape memory effect under IR light irradiation; Video S2. Strength photo-welding joint: Figure S1. TOM micrographs of the self-healing process.

**Author Contributions:** Methodology, T.E.B.P., D.M., J.P., F.I.A. and C.E.H.; Software, J.P., F.I.A. and C.E.H.; Formal analysis, T.E.B.P., J.P., F.I.A. and C.E.H.; Investigation, T.E.B.P., D.M., J.P., F.I.A., T.D.R. and C.E.H.; Resources, J.P., T.D.R. and C.E.H.; Data curation, D.M., J.P., F.I.A. and C.E.H.; Writing—original draft, J.P., F.I.A. and C.E.H.; Writing—review & editing, J.P., F.I.A., T.D.R. and C.E.H. All authors have read and agreed to the published version of the manuscript.

**Funding:** This project has received funding from the European Union's Horizon 2020 research and innovation programme under the Marie-Sklodowska-Curie grant agreement No 101008237. The authors thank the funding provided by CONICET (PIP N° 0594 and 0760 and PIBAA N° 0274), ANPCyT (PICT-18-2309 and PICT21-0578) and The University of Mar del Plata (15/G592).

**Data Availability Statement:** Data are contained within the article and Supplementary Materials.

**Acknowledgments:** T. E. B.-P. thanks the PhD fellowship from CONICET. The help of Gustavo Arenas in the measurements of power irradiances of the NIR sources is gratefully acknowledged.

**Conflicts of Interest:** The authors declare no conflict of interest.

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
