# Peer review of "Synthesis and Processing of Near Infrared—Activated Vitrimer Nanocomposite Films Modified with β-Hydroxyester-Functionalized Multi-Walled Carbon Nanotubes"

_carbon, 2023_

Round 1

Reviewer 1 Report

Comments and Suggestions for Authors

Τhe authors in this article describe the preparation and the characterization of nanocomposite vitrimer films using a specific polymer and functionalized carbon nanotubes. The nanotubes are responsible for the response in the NIR light and the nanocomposite showed very interesting results. The manuscript is well written and documented and I suggest the publication after minor revision.

My only comment is that I think that the introduction and the results and discussion part are quite long. I suggest to the authors to remove detailed descriptions from the results part and the introduction.

Author Response

We thank the reviewer for the suggestion. We revised the introduction and removed some parts to make it shorter and clearer (sections removed are highlighted in the highlighted version of the manuscript).

Reviewer 2 Report

Comments and Suggestions for Authors

It is necessary to find and perform a real application with the material. For example, cooling a determined area or something applied to real world.

Improve quality of the figure captions and figures (i.e. Fig 16)

Comments on the Quality of English Language

Minor revisions in English, specially in the answers to the reviewers

Author Response

It is necessary to find and perform a real application with the material. For example, cooling a determined area or something applied to real world.

We thank the reviewer for the suggestion. In the revised version of the manuscript (introduction section), we highlighted some specific applications that can be envisioned for these films and coatings and included updated references in the topic. Applications for these materials are countless, especially if modifiers are added for conferring them with remote activation. We have only mentioned some of them that can be predicted for the near future as described in the following paragraph: 

“Hence, the advantages and special properties of CANs could be potentially transferred to a new generation of paints, coatings and films for enhancing their durability, versatility and functionality while diminishing their environmental impact. Recent examples of applications of CANs in coatings and films could be found in the development of self-healable paintings for the automotive industry,[10] protective coatings for metal parts,[11,12] flexible optical devices, [13] reversible adhesives,[14,15] soft actuators [16,17] and durable, ultra-thin and self-healable hydrophobic coatings.[18]” 

(see references in the revised version of the manuscript).

Improve quality of the figure captions and figures (i.e. Fig 16)

We thank the reviewer for the suggestion. We added the scale bar measurement in the figure caption and improved the quality of the images. (Please see the revised version of the manuscript)

Reviewer 3 Report

Comments and Suggestions for Authors

The authors reported on the synthesis and processing of near infrared-activated vitrimer nanocomposite films using β-hydroxyester functionalized MWCNTs to modify the epoxy-acid vitrimer. The manuscript is well written and therefore I recommend the publication of this manuscript on C upon the following conditions are well addressed:

[1] The manuscript does not provide a detailed mechanism of the curing process of the films of the vitrimer.

[2] The manuscript does not provide a detailed mechanism of the shape memory effect for the nanocomposite vitrimer films.

[3] The effect of welding of two 533 films by irradiation is not obvious shown in theVideo S2.

[4] Some of the scale bars are difficult to see in the figures.

[5] In Figure 12, what is the unit and scale in the y-axis?

[6] In Figure 15, please highlight the healing region.

Comments on the Quality of English Language

 Minor editing of English language required

Reviewer 4 Report

Comments and Suggestions for Authors

I appreciate the work done by Tomas Prudente and co-workers.

I would like to suggest small improvments:

(i) In the expression “Thermal Oxide Silicon Wafers, Nanovision, 100 nm diameter” it is 100 nm or 100 mm (4 inch?)

(ii) In Fig 1, please indicate with dashed circles or arrows the other peaks correspondence (besides the epoxy, carboxylic acid, hydroxyl groups, indicated in inset graphs). This will help the readers to easily follow the figure

(iii) If possible, please add at least an example of a practical application of the work done ( a bit more concrete than "opening the way to the development of smart and high-performance devices for multiple technological applications")

Round 2

Reviewer 1 Report

Comments and Suggestions for Authors

The manuscript has been improved after the revision of the authors according to the suggestion of the reviewers and I believe that it can be published in the present form. 

Author Response

We are grateful to the reviewer for the suggestions that helped in improving our manuscript.

Reviewer 2 Report

Comments and Suggestions for Authors

I consider this article ready for publication after ahuthors include this answer in the last conclusion paragraph or in the discussion:

"We also demonstrate that the high photothermal effect when dispersed in the of these carbon nanostructures films and can be can be efficiently used for the activation of the shape memory, self-healing, welding and reprocessing of the materials under NIR radiation"

Write in few sentences where you consider in you research that each of the performances you mention (shape memory, self-healing, welding and reprocessing) is achieved.

Author Response

We thank the reviewer once again for the suggestion. 
The activation of the shape memory effect is shown in Video S1 and Figure 14a, and is described in the section “Photothermal Effect: NIR activation of welding, self-healing and shape memory” (starting in page 17). Self-healing and welding were demonstrated in the experiments shown in Figures 14b, 15 and 16, and the corresponding explanations are provided in the same section. 
Regarding the reprocessing, it was proven through direct heating, conveniently depicted in Figure 17. The self-healing and welding of the nanocomposites are allowed by the same underlying phenomenon that the reprocessing, namely the thermally activated transesterification reactions, The only difference is the heat source used to increase the temperature, but either using a convection oven or NIR irradiation would not change the result. To prevent misunderstandings and ease the reading of the article, we removed the word “reprocessing” from that paragraph in the introduction section.

Reviewer 3 Report

Comments and Suggestions for Authors

I think it can be accepted in the presence form.

Author Response

(The authors gave the same response as above.)
